# Event Extraction from Unstructured Amharic Text

## Abstract

In information extraction, event extraction is one of the types that extract the specific knowledge of certain incidents from texts. Event extraction has been done on different languages texts but not on one of the Semitic language Amharic. In this study, we present a system that extracts an event from unstructured Amharic text. The system has designed by the integration of supervised machine learning and rule-based approaches together. We call it a hybrid system. The model from the supervised machine learning detects events from the text, then, handcrafted rules and the rule-based rules extract the event from the text. The hybrid system has compared with the standalone rule-based method that is well known for event extraction. The study has shown that the hybrid system has outperformed the standalone rule-based method. For the event extraction, we have been extracting event arguments. Event arguments identify event triggering words or phrases that clearly express the occurrence of the event. The event argument attributes can be verbs, nouns, occasionally adjectives such as ሰርግ/wedding and time as well.

## 1 Introduction

Amharic is a Semitic language, related to Hebrew, Arabic, and Syriac. Next to Arabic, it has been the second most spoken Semitic language with around 27 million speakers Mulugeta and Gasser (2012) primarily in Ethiopia. It is currently the official language of government in Ethiopia, and has been since the 13th century. It has been the medium of instruction in primary and secondary schools as well as the source language for a large body of historical text. As a result, most documents in the country have been produced in Amharic and there has been an enormous production of electronic and online accessible Amharic documents.

The predominant problems of under-represented languages; There has been lack of resources Sohail and Elahi (2018) for understanding and extracting relevant information from unstructured text. Leading to fewer online resources available to people in their everyday lives and a lack of access to Amharic source texts for scholars and other interested group of people; Now a days, researchers in linguistic and computing disciplines face difficulties because of Amharic presents sophisticated language-specific issues. Thus, any information extraction systems developed for Hebrew, Arabic, or other languages cannot address Amharic problems. Events in Amharic text are predominantly expressed through verbs and nouns, but, the linguistic structure and morphological richness has not represented in models that have been developed for other languages. Events in Amharic are expressed predominantly through verbs and nominal, but, the linguistic structure and morphological richness highly matter to apply models used for other languages.

For example, consider the following sentence "ሰኞ መስከረም 1965 ኢትዮጵያ በውጥረት ነግሳ ነበር." /"Ethiopia was in turmoil in Monday, September , 1965". In this sentence "በውጥረት" and "ነግሳ" refers to an event, whereas the phrase "ሰኞ መስከረም" 1965 is a time argument which indicates when the event happened. The word"ኢትዮጵያ" refers the named entity or participant of the event.

Because of its prominent significance of extracting events from unstructured Amharic text for high level Natural Language Processing (NLP) tasks such as textual entailment, question answering, character identification, semantic role labeling and others we are interested to tackle this problem. In this study we present a comprehensive technique for extracting events from Amharic unstructured text.

## 2 RELATED WORK

Recently event extraction has gained popularity due to its wide applicability for various NLP applications. Most event extraction systems support English and European language texts from different domains using a variety of techniques. Now days, semitic languages are typically a topic of interest for researchers. Event extraction for Amharic has not been done yet; Therefore this study is the first in this particular information extraction (IE) application. Due to the variation of the language structure the existing techniques and tools applied to other languages can't be directly used for this particular task.

There are some progressive works that has been done so far on Amharic natural language analysis tasks with promising results including part of speech tagging, morphological analyzer, named entity recognition, base phrase chunking and text classification as in Adafre (2005); Ibrahim and Assabie (2014); Sikdar and Gambäck (2018); lasker et al. (2007). Various techniques have been widely employed for each task to enhance the accuracy and handling linguistic exceptions. However, there have not been ready-made pre-components and well organized datasets. Besides these limitations there has not been any undergoing research on event extraction from unstructured Amharic text due to difficulties in syntactic and semantic status of class of functional verbs. The other challenges are identifying event arguments. In our case temporal event arguments have considered. The challenge is that temporal expressions in Amharic have represented in various forms such as; Sequence of words, Arabic and Geez'e script numerals. As such it needs extra normalization and syntactic analyzing scheme to tackle temporal argument.

Semitic languages like Arabic, Hebrew and Amharic have much more complex morphology than English. The morphological variation limits the research progress on Natural language processing in general and a very limited works in event extraction task. However, relative to other Semitic languages there are studies as in Al-Smadi and Qawasmeh (2016) which has done for Arabic language on automatic event extraction using knowledge driven approach which concentrates on tagging the event trigger instances and related entities. There has been one great contribution in Al-Smadi and Qawasmeh (2016) which links event to the entity mention. However, in our case we mainly concentrate on extracting events and its arguments with the advantage of hand crafted rules and machine learning classifiers.

Hindi is another under resourced an indo European language, which has more common words with Arabic. In Ramrakhiyani and Majumder (2015) solely focused on Temporal Expression Recognition in Hindi using interactive handcrafted rules. Ramrakhiyani and Majumder (2015) aims to carry out two basic goals, identification of the temporal expressions in plain text and classifying the identified temporal expression. However, extracting events along with the corresponding arguments gains more advantage for the ease of chronological ordering of events in their occurrences. In addition it can be extended for event argument relationship extraction tasks.

Smadi and Qawasmeh (2018) proposed a state-of-the-art supervised machine learning approach for extracting events out of Arabic tweets. This paper mainly focuses on four main tasks: Event Trigger Extraction, Event Time Expression Extraction, Event Type Identification, and Temporal Resolution for ontology population. Significant scores have resulted for each task covered under this paper includes; T1: event trigger extraction F-1= 92.6, and T2: event time expression extraction F-1= 92.8 in T3: event type identification Accuracy= 80.1. Smadi and Qawasmeh (2018) claim that the third task is relatively better than previous works done using similar techniques.

Another work proposed by Arnulphy et al. (2015) detects French and English Time Markup Language(ML) Events by using a combination of different supervised machine learning algorithms such as conditional random field, decision tree and k-nearest neighbor including language models.

Al-Smadi and Qawasmeh (2016) has proposed knowledge-based approach for event extraction from Arabic Tweets. There are three subtasks covered under their study such as event trigger extraction, event time extraction, and event type identification. The event expression includes important event arguments, which are event agent, event location, event trigger, event target, and event product and event time. The tools and dataset used in their study have utilized twitter streaming API and preprocessed through AraNLP Java-based package. Moreover, after the visualization services event extraction like calendar, timeline supplied through the help of ontological knowledge bases.In their study the experimental results show that the approach has an accuracy of, 75.9 for T1: event trigger

extraction, 87.5 for T2: Event time extraction and 97.7 for T3: event type identification. Al-Smadi and Qawasmeh (2016) claims that applying this kind of domain dependent approach to extract events from tweets scores significant results.

In general there has been a lot of work in event extraction such as Arnulphy et al. (2015); Tourille et al. (2017) in European languages, predominantly English, there has been much less research in other languages. Extracting events in one language may apply to languages with a similar grammar and character set, unlike to apply to languages with a very different grammar, or a very different written representation. There has been research in part-of-speech tagging on Amharic text Adafre (2005) and on Amharic morphology Mulugeta and Gasser (2012) which are helpful for event detection, but not directly related where state of the art Event detection typically uses a robust machine-learning techniques. Examples of such systems are Arnulphy et al. (2015). Because of the lack of sufficient labeled training data for Amharic, we bootstrap an event extractor using a rule-based algorithm.

## 3 METHODOLOGY

According to Frederik Hogenboom and Kaymak (2016) event extraction techniques have been evaluated based on the works on a set of qualitative dimensions, i.e., the amount of required data, knowledge,expertise,interpretability of the results and the required development and execution times. In this study, supervised machine learning techniques, handcrafted rules and hybrid techniques has employed to detect and extract events and its arguments from unstructured text. Our focus of interest has been extracting events and event arguments from unstructured Amharic text. Event arguments include identification of event trigger words; Where in Amharic unstructured text nominal events become ambiguous. Such events can be arguments of other events, and they often have been hard to be identified.

### 3.1 DATA SET PREPARATION

Unlike other languages, Amharic language does not have any standardized annotated publically available corpora like Treebank and propbank for English. The news domain has been preferable data source because of its publically availability and rich source of information for any NLP applications such as entity extraction, event and temporal information extraction and co-reference resolution. In this study we build our own data set by scraping top websites Zehabesha[1], Satenaw[2] , Ezega[3] ,and BBC Amharic[4] which contains relevant Amharic unstructured text contents. A Python Beautiful Soup library [5]has been used for pulling data out of HTML and XML files. The scraped texts are from all domains such as economy, politics, technology and sport. Simple regular expressions have been used to retrieve only relevant text contents; A total of 659,848,657 words have extracted. Along with our own dataset we have used Amharic corpora which have been prepared by the Ethiopian Languages Research Center of Addis Ababa University in a project called *the annotation of Amharic news documents*. The project has been tagging manually each Amharic word in its context with the most appropriate parts-of speech. The corpus has 210,000 words collected from 1065 Amharic news documents of Walta Information Center Demeke and Getachew (2006), a private news and information service located in Addis Ababa.

### 3.2 DATA PREPROCESSING

In this step, data has converted to the appropriate format required for the respective information extraction process. In this study the scraped texts have many junks including markup tags and other special characters. The first step in our study is raw text preprocessing. The raw text preprocessing in this research includes cleaning unwanted junks, sentence splitting, tokenizing, word stemming, character normalization, stop word removal and Part Of Speech tagging (POS). Unlike other languages, Amharic is a morphologically rich language which posses complicated syntactic features.

---

[1]http://www.zehabesha.com/amharic/

[2]https://www.satenaw.com/amharic/

[3]https://www.ezega.com/News/am/

[4]https://www.bbc.com/amharic

[5]https://www.crummy.com/software/BeautifulSoup/bs4/doc/

This makes cumbersome the preprocessing tasks to analyze the morphological features of representative tokens. The sentence splitter splits using Amharic sentence demarcations (.፤ ? !).

Amharic language has different characters with the same meaning and pronunciation with different symbols. Those different symbols should be treated equally because there is no change in meaning regardless of the linguistic view of orientation among characters. For Example:- ( ሀ፣ሐ፣ኀ) , ( ሰ፣ ሠ) , ( ዐ፣ ዓ፣ አ) and ( ጸ፣ ፀ)  have the same meaning Gasser (2011). As a result, we develop a character normalizer which enables to normalize those characters to an ordinary conceivable form. This task helps the performance of our system. The other preprocessing task is stop word removal ,like any other language Amharic has its own list of stop words including conjunctions, articles and prepositions. In our case we have adopted stop word lists used in Tsedalu (2010) and we build our own stop word lists with the help of linguistic experts. Then a total of 235 stop word lists have identified.

The other important preprocessing task is analyzing Amharic verb morphology to identify lemma of words and their derivation. The lemma of a word is very crucial feature for the classifier. We applied hornmorpho [6] that is a system for morphological processing of Amharic, Affan Oromo, and Tigrinya which are Ethiopian local languages. The hornmorpho misses some unique and compound words thus we have developed our own unique exceptional dictionary (Gazetteer) to handle exceptional keywords. Finding a pattern to get only the lemma of the hornmorpho result has also other difficulties; Because sometimes the hornmorpho skips subject, object, grammar, or word classes of a specific words; If the word doesn't contain full information. For Amharic Language the evaluation of Hornmorpho has conducted in Gasser (2011) using 200 randomly selected verbs and nouns/adjectives. The output compared with manually identified Amharic verbs: 99%; Amharic nouns: 95.5%. Because of lack of other readymade NLP components for Amharic we preferred to use this tool in our study. The Jython library([7]) has used to integrate the python based morphological analyzer for Amharic to get morphological features of words.

Besides analyzing the verb morphology annotating the exact word classes of the instance is also the required preprocessing task in this study. To do so, we have been using the publically available language independent part-of-speech tagger, which is TreeTagger[8], has been used to annotate Amharic texts with their proper part-of-speech. TreeTagger is a tool for annotating text with part-of-speech and lemma information. The TreeTagger has been successfully used to tag German, English, French, Italian, Danish, Swedish, Norwegian, Dutch, Spanish, Bulgarian, Coptic and Spanish texts and has been adaptable to other languages if a lexicon and a manually tagged training corpus are available Schmid (1994).The TreeTagger consists of two programs: the training program creates a parameter file from a full-form lexicon and a hand tagged corpus. The tagger program reads the parameter file and annotates the text with part of speech and lemma information. To prepare a parameter file for TreeTagger we used a total of 217,000 Amharic manually tagged corpora with 9 distinguished word classes and corresponding lemmas. We have conducted evaluation of TreeTager using 92,456 randomly untagged tokens. The output compared with manually tagged Amharic words which results 99.9% accuracy. Another crucial step in our preprocessing module is normalizing Amharic temporal arguments. There have been various representations of date time expressions in Amharic including Arabic, Geez and using alphanumeric characters.

For Example consider the following sentences;

(አቤል በ1995 ዓ.ም ተወለደ ።)
(አቤል በሺ ዘጠኝ መቶ ዘጠና አምስት ዓ.ም ተወለደ ።)
(አቤል ስኚ መስከረም 1995 ዓ.ም ተወለደ ።)
(አቤል ስኚ ጠዋት መስከረም 1995 ዓ.ም ተወለደ ።)
 All the above sentences refer logically similar meaning with various syntactic representation. In order to handle temporal arguments of the event, a normalization and conversion scheme to convert temporal representations into one form. The conversion of Ge'ez numerals to uniform Arabic number system is not straight forward as other normalization tasks because of the irregularities of Unicode values for Ge'ez numerals.

---

[6]https://www.cs.indiana.edu/ gasser/HLTD11/

[7]https://www.jython.org/

[8]https://reckart.github.io/tt4j/

### 3.3 EVENT DETECTION USING SUPERVISED MACHINE LEANING

In this study, supervised machine learning technique has been employed. Supervised machine learning classifiers typically predict new events, based on the given labeled training sets. Such learning algorithms deduce event properties and characteristics from training data and use these to generalize the unseen situations.

In this study, the datasets are unstructured text and documents. However, these unstructured text sequences must be converted into a structured feature space using mathematical modeling. Feature extraction for classification can be seen as a search among all possible transformations of the feature set for the best one. This preserves class reparability as much as possible in the space with the lowest possible dimensionality. Features have properties of a text that have used to provide necessary information associated to a given events and increase the confidence level of predicting a token as an event. Thus, in this study the feature extractor component is responsible for extracting candidate attributes for the classifier. The features that have used in this study are the following:-

- Words of the instance
- POS of the corresponding word
- Lemma of the corresponding word
- List of lexicons for exceptional events

A binary classifier has been used to detect events from Amharic text. The classifier detects events from the text and classify the text as on-event and off-event. The on-event class represents the instance which contains event trigger keywords ; Whereas the off-event class refers the instance which is do not infer the event trigger keywords.

From the machine learning algorithms Naive Bayes,decision and SVM algorithms have been selected based on their widely use in text classification task such as Pranckevicius and Marcinkevicius (2017); Bilal and Israr (2016); Sarkar and Chatterjee (2015)

Naive Bayes classifier is linear classifier that has been known for being simple and very efficient for text classification task. The probabilistic model of naive Bayes classifiers is based on Bayes' theorem. This algorithm works on the assumption that the features in a dataset have been mutually independent.

LIBSVM is a library for Support Vector Machines (SVMs). It has gained wide popularity in machine learning and many other areas. SVM finds an optimal solution and maximizes the distance between the hyperplane and the difficult points close to decision boundary. The assumption Chang and Lin (2011)is if there are no points near the decision surface, then there are no very uncertain classification decisions.

The last classifier algorithm that has applied in this study is decision tree. Decision tree is Tree-based classifier for instances represented as feature-vectors. There is one branch for each value of the feature, and leaves specify the category. It represents arbitrary classification function over discrete feature vectors.

The models have trained on above algorithms using the labeled data-set as input. The prediction phase gets new input and detects an event as on-event and off-event classes. The best feature that has been recommended by the system which is more powerful than other feature sets to predict the event classes. The POS has found as the best syntactic feature to detect the events based on the feature selection recommendation.

### 3.4 EVENT EXTRACTION USING RULE BASED APPROACH

The knowledge-driven approach requires more expert knowledge. It works adequately on a small dataset. Rule learning based is one of IE method which utilizes the extraction pattern to retrieve information from a text document. A standalone rule-based approach has proposed to enhance the accuracy of our event extraction system over its coverage. Unlike other languages, Amharic has a subject-object verb agreement and other morphological features makes cumbersome the construction of rules. As Yunita Sari and Zamin (2010) has mentioned, construction of extraction pattern is based on syntactic or semantic constraint and delimiter based or combination of both syntactic and

semantic constraint. Events dominantly exist as nominal and verbs Ramesh and Kumar (2016). The nominal events are ambiguous, in which they can appear in deverbal or non-deverbal nouns form. Thus , to disambiguate nominal events we need morphological features of the instances. To do so, morphological analyzer has employed to get the morphological features of event mention instances. e.g (ምሳሌ)፦ ( የኢትዮጵያ ህዝቦች ከዚህ ቡሀላ ፈፅሞ አምባገነናዊ ስርአት አያስተናግዱም ። ) In this sentence the underline word (ፈፅሞ) is derived from the verb (ፍጹም) it seems an adjective, but, it's a deverbal entity we call it a nominal event. The rules have been developed based on syntactic features of words with help of a carefully constructed list of gazetteers. The word class (POS tag) and lemma of the word itself have been used as a abasement for the handcrafted linguistic rules. Different components have used to get syntactic features of words using Tree Tagger for Java (T4J), Hornmorpho for Amharic, Tigrigna and Affan Oromo. The pattern extractor has been developed based on those syntactic features. Simple rules have been applied to extract detected events. e.g / (ምሳሌ)፦ (አበባ) N (ትላንት) ADV (የገዛው) VN (በሬ) N (ሞተ) VP ( ። ) Here the snippets of handcrafted rules have been tackled based on the POS tagger results. In addition, the formal structures are not always regular to develop stable rules. Whereas the morphological analyzer had been very helpful, because of the existence of deverbal events which have been act as ambiguous.

Some of the rules those have been applied in this system includes the following :-

1. Automatically label preprocessed texts with their corresponding word classes or parts-of-speeches.

2. Get the morphological features of words including; Word, subject,root,cit,object, grammar and preposition

3. Usually events expressed using verbs and nouns. Check the neighboring words using bi-gram language models. Because not all nouns have been events and sometimes nouns come at the beginning are the subjects or participant of the event not exactly the event.

4. Identifying the nominal events; To do so, the morphological analyzer has great role it indicates the citation of the respective nouns; i.e words which have been exactly nominal can be deverbal or non deverbal nouns, but, deverbal nouns has a citation of verbs.

5. Words which has been categorized as verbs and verb group word classes as part-of-speech and it's infinitive forms have selected as primary candidates.

6. Check non deverbal nouns (usually acts as events) from carefully built gazetteers(List of non deverbal noun lexical). Because of our limited dictionary a ternary search tree algorithm has been applied to enhance the efficiency.

7. Identifying words which contains temporal keywords; Those temporal indicator keywords have been carefully built list of commonly used temporal expressions in Amharic. In addition regular expressions have been constructed to tackle regular date-time expressions. Bi-gram language models have been applied to find temporal arguments.

8. Example የአበበ ሰርግ ነገ ነው ። / "Abebe's wedding is torrow." from the above sentence the word ሰርግ is a deverbal nouns which has been extracted as an event and it's actually an event. where the word ነገ is an event argument extracted as temporal event argument of the major event ሰርግ

## 3.5 EVENT EXTRACTION USING HYBRID APPROACH

In hybrid event extraction systems, due to the usage of supervised machine learning techniques, the amount of required data increases with respect to knowledge-driven systems, yet typically remains less than the case with purely data-driven techniques. Compared to a knowledge-driven techniques, complexity and hence required expertise is generally high due to the combination of multiple techniques.The interpretability of a system benefits to some extent from the use of semantics as in knowledge-based techniquesBaradaran and Mineai-Bidgoli (2015). The last technique that has employed in this study is combining both supervised machine learning and rule-based techniques to extract events from Amharic unstructured text. The power of the machine learning approach has been mainly focused on coverage (Recall) apart from sensitivity (precision). While, the handcrafted rules have been able to achieve the highest potential of precision value based on the incorporated rules. In our case the machine learning classifiers miss nominal events in comparison with the verbal

events. Having such limitation we incorporate some rules to tackle the missed events from the machine learning classifiers result. Deverbal nouns exhibit both nominal and verbal syntactic behavior: they operate as concrete nouns, but also participate in verbal constructions where they require arguments and accept the aspectual modification. Nominal events sometimes appear as deverbal and non-deverbal, in which deverbal entities have been derived from verbs in-contrary the non-deverbal entities have not derived from verbs. e.g. / (ምሳሌ) (ፈጽም) is a deverbal entities which is an event derived from verb (ፍጹም) . By definition, an event is a situation which lasts for a moment. Having this definition, nominal can be an event e.g (ሰርግ) wedding is a non-deverbal nominal event. e.g. (የአበበ ሰርግ ሃምሌ 16፤ 2010 ዓ.ም ነው .) Simply knowing the morphological variation of words and having a common non-deverbal nominal list of carefully constructed gazetteers (list of exceptional non deverbal events) helps to get rid of event ambiguity. We also get those deverbal events from the morphological analyzer and non-deverbal events from the gazetteers. Applying such disambiguation scheme improves accuracy of our system in proportion to the standalone rule based approach.

## 4 EXPERIMENTAL RESULTS

In this study, a total of five experiments have been conducted. For each tasks we applied different models. As a result, we have used the dataset in accordance with the model's requirement. The results of the experiments have been evaluated using standard information extraction performance metrics including precision, recall-measure, and ROC (Receiver operating characteristics curves).

Among the standard information extraction evaluation metrics precision, recall and F-measure has been used to evaluate the performance of machine learning classifiers. We applied k-fold cross-validation which split our dataset randomly into k groups. In this case by shuffling the dataset randomly one of the groups is used as the test set and the rest has used as the training set. The model has been trained on the training set and scored on the test set. We have used all features to see the effect of each attribute on the event detection. Each algorithm has been experimented with full features.

In this model among the three algorithms the Naïve Bayes classifier performs better than the other classifiers by correctly classifying 90.11% of the instances correctly. In this study the experimental result confirms the advantage of Naïve Bayes classifier for event detection task. Because of its assumption on the probability of occurrence of any word given the class label is independent of the probability of occurrence of any other word, given that label. Naive Bayes also consider the probability of occurrence of a word in a document, is independent of the location of that word within the document. The following table shows the experimental result for the machine learning classifiers. We get encouraging result using a machine learning classifier for event detection task.

Table 1: Experimental results for machine learning Algorithms to detect events

| Algorithms | Measures | | | Classes |
|---|---|---|---|---|
| | *Precision* | *Recall* | *F-measure* | |
| **NB** | 0.0.866 | 0.798 | 0.831 | **ON-Event** |
| | 0.932 | 0.957 | 0.944 | **OF-Event** |
| | 0.915 | 0.916 | 0.915 | **Weighted Ave.** |
| **LIBSVM** | 0.895 | 0.395 | 0.548 | **ON-Event** |
| | 0.825 | 0.984 | 0.897 | **OF-Event** |
| | 0.843 | 0.833 | 0.808 | **Weighted Ave.** |
| **J48** | 0.891 | 0.698 | 0.783 | **ON-Event** |
| | 0.903 | 0.971 | 0.935 | **OF-Event** |
| | 0.9 | 0.9 | 0.896 | **Weighted Ave.** |

The problem resides on deverbal entities ambiguousness. Next to this our standalone rule based event extraction system has been evaluated using standard metrics. From the machine learning based event detection system we observed that due to linguistic features verb trigged events has get equal weight by the classifier with that of non-event class. This has been the reason which motivates us to come up with developing hand crafted rules to get rid of the ambiguities. In this particular

technique in order to make a clear comparison with the hybrid based event extraction system we have used similar data set.

The last experiment that has been conducted in this study is the hybrid event extraction technique. The performance of this method relay on the power of having the advantage of the rule based and supervised machine learning methods in conjunction. As we said earlier to make a fair judgment on the performance of each technique we have been using the same data set. The machine learning classifier label instances as on-event and off-event binary classes by assigning different weights. An instance which gets highest probability to be an event is the one which has been categorized as on-event class by the classifier. In the other case the off-event class instances have been mostly non-event. We accept positive predicated values as it's i.e. instances categorized as on-event with highest weighted value. Because, it is predicted exactly as an event, while instances getting equal weight by the classifier in both class are going to be the target instances for the heuristics. Using the help of syntactic features ambiguous instances has been handled. As a result, the number of event instances extracted increases when heuristics has applied. In order to get the false negative and the false positive we have used a manual scanning of the result to be accurate.

The following table shows the comparison of experimental results of rule based and hybrid techniques to extract events from unstructured Amharic text. From the table below we realize that the combination of both rule based and supervised machine learning classifiers bring significant result to extract events from unstructured Amharic text.

Table 2: Over all event extraction evaluation of experimental result comparison

| Techniques | Standard measures | | |
|---|---|---|---|
| | *Precision* | *Recall* | *F-measure* |
| Rule based Approach | 0.976 | 0.952 | 0.959 |
| Hybrid Approach | 0.979 | 0.962 | 0.971 |

## 5 CONCLUSION AND FUTURE WORK

In this study we presented on event detection and extraction from unstructured Amharic text. Supervised machine learning classifiers such naïve bayes, support vector machine and decision tree have used to detect events. Stand alone rule based approach and hybrid approach have also employed to extract events. Our system has been evaluated on our own data-set using standard evaluation metrics precision, recall and F-measure. From the study we have showed that the hybrid approach have been informative, and outperformed the standalone rule based approach to extract events from the text.

In the future we need to address other relevant event extraction tasks such as; Build larger event and temporally annotated corpus, employing powerful deep learning techniques to extract relation between event and time, extracting relation between events and document creation time.

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
