# OpenReview forum: "Event extraction from unstructured Amharic text"
_ICLR.cc/2020/Conference — Reject_

### Official Review · AnonReviewer3 · 2019-10-20
**Official Blind Review #3**

**Rating:** 1

**Review:**

The paper describes a dataset an a hybrid machine learning + rule-based approach for event extraction from Amharic Text.

Although there has been a lot of work on event extraction for other languages, including morphologically rich ones, this paper states there has not been prior work on event extraction in Amharic.

The authors describe a new event extraction dataset, text for which they scraped from the news websites. The raw text collection is large, with 659 million words. It is not clear to me how the manual annotation for events was done, how many event types there are; I did not see a statement of inter-annotator agreement. According to Table 1, there seem to be only two classes: has event and does not have an event, but this was not clearly stated.

I think the paper in its current form should be rejected from ICLR. Overall the approaches in this work are reasonable but not new. The paper is not clearly written.

A new manually annotated corpus in Amharic is a good contribution. My suggestion is to submit to another venue such as LREC, focus on the new resource and include linguistic insight into the language-specific challenges, and also include machine-learned and rule-based baselines. A more modern baseline including unsupervised representation learning (i.e. multi-lingual pre-trained contextual representations) would be necessary as well.


**Experience Assessment:**

I have published in this field for several years.

**Review Assessment: Checking Correctness Of Derivations And Theory:**

N/A

**Review Assessment: Checking Correctness Of Experiments:**

I assessed the sensibility of the experiments.

**Review Assessment: Thoroughness In Paper Reading:**

I read the paper at least twice and used my best judgement in assessing the paper.

---

### Official Review · AnonReviewer1 · 2019-10-23
**Official Blind Review #1**

**Rating:** 1

**Review:**

This paper deals with the important issue of information extraction from less-resourced languages.
In its current form, the paper has two main weaknesses:
- it is poorly written & organized
- it was a fairly weak empirical evaluation

In order to address the first issue:
- the authors should significantly improve the quality of the prose, which can be confusing & difficult to undersrand
- the introduction needs to be significantly crisper: in its current form, it is far too general and does NOT describe the rest of the paper; the authors should explain ...
   1) what is the problem they are working on (currently present, but far too long)
   2) what is the proposed approach & why is it novel (missing)
   3) what are the main results & their significance (also missing)

In order to address the second issue:
- 3.1 needs more details; it is this reviewer's understanding that the current corpus consists of 1065 documents (which is extremely small in size); how many sentences are there in these documents? how many on/off events?
- 3.2 is particularly difficult to follow
- 3.3 should not use 3-4 paragraphs on introducing NB, SVM, and Decision Trees; last but not least, why decision trees and not randomized forests or deep learning models? in the current form, it is also unclear whether the on/off event detection is performed at sentence or document level
- 4: what is the value of k in k-fold CV?

Last but not least, the authors could use the work below as inspiration on how to improve the overall quality of the paper
https://pqdtopen.proquest.com/doc/2025917601.html?FMT=ABS

**Experience Assessment:**

I have published one or two papers in this area.

**Review Assessment: Checking Correctness Of Derivations And Theory:**

I assessed the sensibility of the derivations and theory.

**Review Assessment: Checking Correctness Of Experiments:**

I carefully checked the experiments.

**Review Assessment: Thoroughness In Paper Reading:**

I read the paper thoroughly.

---

### Official Review · AnonReviewer2 · 2019-10-26
**Official Blind Review #2**

**Rating:** 3

**Review:**

The paper studies extracting events from unstructured text, specifically on the low resource language Amharic. The paper proposes to combine rule-based based with a learning-based approach.

Strength
-	Looks at the low resource language Amharic and mines a large corpus of text.
-	The paper compares 3 learning algorithms, NB, Libsvm, J48. As well as a hybrid versus a rule based approach.

Weaknesses:
1.	Prior work
1.1.	The paper misses to discuss prior work on Amahric copora
Rychlý P., Suchomel V. (2016) Annotated Amharic Corpora. In: Sojka P., Horák A., Kopeček I., Pala K. (eds) Text, Speech, and Dialogue. TSD 2016. Lecture Notes in Computer Science, vol 9924. Springer, Cham
2.	Experimental evaluation
2.1.	Missing ablations
2.1.1.	The paper claims “Amharic presents sophisticated language-specific issues” but does not evaluate how the proposed approach handles those specific challenges.
2.1.2.	The authors “normalize” characters with the same meaning “to an ordinary conceivable form” which “helps the performance of [their] system”; however, the paper does not ablate this benefit.
2.2.	The authors “applied k-fold cross- validation which split our dataset randomly into k groups”. It would be great if the authors report the standard deviation across the results to understand the significant of differences.
2.3.	The paper misses to show any qualitative results showing how the system works and which error it makes.
2.4.	The paper notes that the “Amharic has a subject-object verb agreement and other morphological feature makes cumbersome the construction of rule” as well as the large unsupervised corpus they have available suggest to me that the authors should explore unsupervised pre-training a transformer-based network, such as BERT or other approaches which have successfully been used on other languages.
3.	Clarity: The paper’s clarity could be improved:
3.1.	On the one hand by clearly stating the contributions and by improving the clarity of the writing throughout.
3.2.	It is not clear how Table 2 relates Table 1. The authors note it is the same dataset. Does Table 2 show Weighted Ave of On and Off-event?
3.3.	What is the large unlabeled corpus used for?

Minor:
- considering using the correct citation style: “27 million speakers Mulugeta and Gasser (2012) “ -> 27 million speakers (Mulugeta and Gasser, 2012)
- don’t capitilzie after “;”:
- duplication of very similar sentences: “Events in Amharic text are predominantly expressed through verbs and nouns, but, the linguistic structure and morphological richness has not represented in models that have been developed for other languages. Events in Amharic are ex- pressed predominantly through verbs and nominal, but, the linguistic structure and morphological richness highly matter to apply models used for other languages.”

Conclusion:
While it is interesting and valuable that this paper studies a low-resource language. The paper misses comparisons to prior work for event extraction and misses to include ablations which show the value of the introduced rules and design decisions made.

=== post author response ====
I thank the authors for their response.

Unfortunately, the authors did not use the chance to update the pdf to improve the clarity (3.); report standard deviation (2.2) to judge significance, although they should have this data; nor did they include a revised discussion of prior work.

As clarity and experimental evaluation are also major concerns of the other reviewers and it is unclear if and how they will be addressed in a revision I do not recommend accepting the paper.


**Experience Assessment:**

I do not know much about this area.

**Review Assessment: Checking Correctness Of Derivations And Theory:**

N/A

**Review Assessment: Checking Correctness Of Experiments:**

I assessed the sensibility of the experiments.

**Review Assessment: Thoroughness In Paper Reading:**

I read the paper at least twice and used my best judgement in assessing the paper.

---

### Decision · Program_Chairs · 2019-12-19

**Decision:**

Reject

**Comment:**

This paper performs event extraction from Amharic texts. To this end, authors prepared a novel Amharic corpus and used a hybrid system of rule-based and learning-based systems.
Overall, while all reviewers admit the importance of addressing low-resource language and the value of the novel Amharic corpus, they are not satisfied with the quality of the current paper as a scientific work.
Most importantly, although the attempt of even extraction might be new on Amharic, there have been many works on other languages. It should be clearly presented what are the non-trivial language-specific challenges on Amharic and how they are solved, otherwise it seems just an engineering of existing techniques on a new dataset. Also, all reviewers are fairly concerned about the presentation and clarity of the paper. Unfortunately, no revised paper is uploaded and we cannot confirm how authors' response is reflected. For those reasons, I would like to recommend rejection.